# Variable Neighborhood Search Algorithms for an Integrated Manufacturing and Batch Delivery Scheduling Minimizing Total Tardiness

**Cheol Min Joo [1] and Byung Soo Kim [2,*]**

[1] Division of Mechatronic Engineering, Dongseo University, Busan 617-716, Korea; cmjoo@dongseo.ac.kr
[2] Department of Industrial and Management Engineering, Incheon National University, Incheon 22012, Korea
* Correspondence: bskim@inu.ac.kr; Tel.: +82-32-835-8482

**Abstract:** This article addresses an integrated problem of one batching and two scheduling decisions between a manufacturing plant and multi-delivery sites. In this problem, two scheduling problems and one batching problem must be simultaneously determined. In the manufacturing plant, jobs ordered by multiple customers are first manufactured by one of the machines in the plant. They are grouped to the same delivery place and delivered to the corresponding customers using a set of delivery trucks within a limited capacity. For the optimal solution, a mixed integer linear programming model is developed and two variable neighborhood search algorithms employing different probabilistic schemes. We tested the proposed algorithms to compare the performance and conclude that the variable neighborhood search algorithm with dynamic case selection probability finds better solutions in reasonable computing times compared with the variable neighborhood search algorithm with static case selection probability and genetic algorithms based on the test results.

**Keywords:** variable neighborhood search; meta-heuristic; scheduling; total tardiness; mixed integer programming

## 1. Introduction

Over the past several decades, supply chain management (SCM) has emerged as an important topic and many operational research problems for the SCM have been attracted. Even though many researchers obtained local operational efficiency by optimizing the logistics flow of each entity (i.e., raw-material providers, manufacturing plants, whole-sale distributers, and customers) within supply chain (SC), they have become to recognize that the coordination between the entities in the SC is important to obtain global efficiency of logistics throughout the entities in the SC. Due to this reason, an integrated schedule of manufacturing and delivery has recently received great attention from the researchers.

One review article by Chen [1] introduced several single-period optimization models for integrating inbound-production and outbound-truck scheduling in the SC. The article introduced the integrated optimization models based on various objective function types using time, cost, and profit. For integrated scheduling problems between production and delivery, Fan et al. [2] studied the integrated production with a single machine and delivery scheduling with batching. The limitation of the study was that they considered the batch delivery to only one customer. Cakici et al. [3,4] investigated a similar problem with Fan et al. [2]. They extended the integrated scheduling problem to parallel machines in a production plant and multi-customers for batch delivery. However, they assumed that the delivery operation was processed using only a single truck. Agnetis et al. [5] studied the coordination problem of the batching and delivery problem, where product-part batches were

delivered between production sites by a 3PL provider. Between the delivery process from upstream to downstream production sites, only the batch with all jobs completed at the upstream site can be delivered to the downstream site with two transportation modes. Li et al. [6] studied a coordination problem between the assembly manufacturing plant with parallel machines and multi-destination transportation with a constraint of make to order (MTO) inventory strategy. They decomposed the overall problem into a sub-problem of parallel machine scheduling and a sub-problem of 3PL transportation and solved the problem. Chang et al. [7] considered a coordination problem between a manufacturing plant with the unrelated parallel machines and multi-destination transportation with capacitated delivery trucks. Their problem minimizes biobjectives with the total distribution cost and the delivery time without batching and inventory strategies. Li et al. [8] considered a coordination problem of vehicle schedule and routing between the manufacturing plant with the parallel batch machines and multi-destination transportation with capacitated delivery trucks by the third-party logistics provider. The objective function of the article was to maximize the total profit of the company.

Meanwhile, an integrated scheduling problem between two production processes had a structural similarity with the integrated scheduling problems between production and delivery in the SC. Several studies on the two-stage production and assembly scheduling problem were introduced [9–11]. In recent years, several meta-heuristic algorithms were proposed to optimally solve the integrated scheduling problem between production and delivery under different problem frameworks [12–18].

The fulfillment of due dates of customers is important to obtain a global logistics efficiency in the overall supply chain. Even though the tardiness factor is a significant factor on the integrated scheduling, to the best of our knowledge, a few meta-heuristic algorithms generate a near-optimal solution for the integrated scheduling problems. Furthermore, to the best of our knowledge, none of the research has an integrated scheduling problem, including a batching decision between the scheduling problems with a tardiness objective measure. In this article, based on the contribution of the problem, we propose two effective and efficient variable neighborhood search (VNS) algorithms for minimizing total tardiness of our integrated scheduling framework.

## 2. Problem Statement and Mixed Integer Linear Programming (MILP) Model

A number of orders were sequentially carried out by manufacturing, batching, and delivery operations between a manufacturing plant and multi-delivery sites. Many jobs in the various orders by customers were firstly received and produced by one of the identical parallel machines in a manufacturing plant. The jobs to be shipped to the same customer were grouped in a batch. The batch was loaded into one of the available trucks with a truck containing limit and delivered to the associated customer. Once the trucks were successfully delivered to the current delivery location, for the next delivery, they were directly returned to the manufacturing plant. In this article, three main decisions are to be determined: (1) machine scheduling, which gives a job assignment to a machine and a job sequence produced in each machine, (2) batching, which decides grouping the jobs to the same delivery place within a delivery capacity, and (3) truck delivery scheduling, which decides a batch assignment to truck and a batch sequence delivered in each truck. Total tardiness violating due times of each job is important to improve the service level of customers. Thus, the objective function is to minimize the total tardiness. For the mathematical formulation, the parameters and decision variables were defined in Appendix A.

In this model, the variables $z^M_{iim}$ and $z^T_{kkt}$ were specially introduced. The variable $z^M_{iim}$ equals 1, if job $i$ is assigned to the first processing sequence of machine $m$ at the manufacturing plant. Similarly, the variable $z^T_{kkt}$ equals 1, if batch $k$ is assigned to the first delivery sequence to truck $t$. Since the batching is one main decision in the model, the initial set $B$ is defined as the set of maximum available batches and the number of maximum available batches equals the number of jobs ($|B| = |J|$). Some dummy batches had no assigned jobs in set $B$, and we ignored the batches on the truck delivery scheduling. In these cases, we ignored the batches on the truck delivery scheduling, which were $y^C_{kn} = 0$ for $\forall n \in C$.

For illustrating the proposed problem, a simple example is given in Figure 1 and Table 1. In the example, nine jobs from three orders by the corresponding customers were required to schedule manufacturing and delivery by two machines and two trucks. Jobs 1 and 2 were ordered by customer 1, jobs 3, 4, 5, and 6 were ordered by customer 2, and jobs 7, 8, and 9 were ordered by customer 3, respectively. In Table 1, the parameters of processing time, due time, and volume of each jobs, and transportation time to each customer are shown. From Figure 1, machine 1 sequentially produces jobs 3, 7, 5, and 9, and machine 2 also sequentially produces jobs 1, 4, 8, 6, and 2 at the plant. According to the truck containing capacity ($V = 10$) and the ordering customer, 6 batches were grouped using the manufactured jobs. Once batching was completed, truck scheduling was processed based on the batches. For truck scheduling, truck 1 sequentially delivered three batches 3, 5, and 6 to customers 2, 3, and 3, and truck 2 sequentially delivered three batches, 1, 4, and 2, to customers 1, 2, and 1, respectively. From these schedules, the tardiness of each job is calculated in the last column of Table 1. Hence, the total tardiness of these schedules becomes 150.

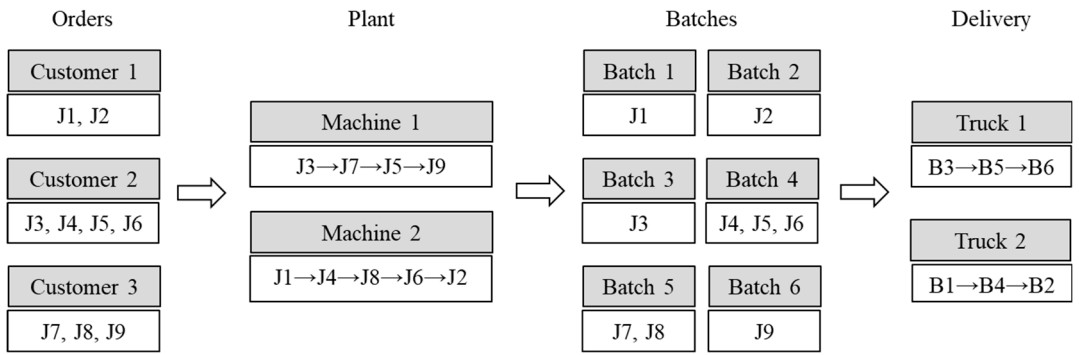

**Figure 1.** An example of schedules for the integrated scheduling problem.

**Table 1.** Input data and the resulting tardiness of each job.

| Orders | | | | | | Plant | | Batches | Delivery | | | Tardiness |
|---|---|---|---|---|---|---|---|---|---|---|---|---|
| Customer | Job | $p_j$ | $d_j$ | $v_j$ | $h_n$ | Machine $x_j$ | $x_j+p_j$ | Batch | Truck | $r_k$ | $r_k+h_n$ | |
| 1 | J1 | 40 | 150 | 8 | 90 | 2 | 0 | 40 | B1 | 2 | 40 | 130 | 0 |
| | J2 | 60 | 300 | 7 | | 2 | 140 | 200 | B2 | 2 | 240 | 330 | 30 |
| 2 | J3 | 30 | 100 | 10 | | 1 | 0 | 30 | B3 | 1 | 30 | 130 | 30 |
| | J4 | 30 | 200 | 3 | 100 | 2 | 40 | 70 | | | | | 40 |
| | J5 | 50 | 250 | 4 | | 1 | 80 | 130 | B4 | 2 | 140 | 240 | 0 |
| | J6 | 20 | 250 | 2 | | 2 | 120 | 140 | | | | | 0 |
| 3 | J7 | 50 | 180 | 5 | | 1 | 30 | 80 | B5 | 1 | 120 | 200 | 20 |
| | J8 | 50 | 200 | 4 | 80 | 2 | 70 | 120 | | | | | 0 |
| | J9 | 40 | 250 | 7 | | 1 | 130 | 170 | B6 | 1 | 200 | 280 | 30 |

Using the above problem parameters and decision variables, the MILP model for the proposed problem is as follows:

$$Min\ z = \sum_{i \in J} \tau_i \qquad (1)$$

s.t.

$$x_i + p_i \le x_j + Q \cdot \left(1 - \sum_{m \in M} z_{ijm}^M\right), \quad for\ \forall i, j \in J;\ j \ne i \qquad (2)$$

$$\sum_{m \in M} y_{im}^M = 1, \quad for\ \forall i \in J \qquad (3)$$

$$\sum_{i \in J} z_{iim}^M \le 1, \quad for\ \forall m \in M \qquad (4)$$

$$\sum_{j\in J} z_{jim}^{M} = y_{im}^{M}, \quad for\ \forall i \in J;\ \forall m \in M \tag{5}$$

$$\sum_{\substack{j\in J \\ j\neq i}} z_{ijm}^{M} \leq y_{im}^{M}, \quad for\ \forall i \in J;\ \forall m \in M \tag{6}$$

$$\sum_{k\in B} y_{ik}^{B} = 1, \quad for\ \forall i \in J \tag{7}$$

$$\sum_{i\in J} v_{i} y_{ik}^{B} \leq V, \quad for\ \forall k \in B \tag{8}$$

$$y_{ik}^{B} + y_{jk}^{B} \leq 1 + \sum_{n\in C} R_{in}\cdot R_{jn}, \quad for\ \forall k \in B;\ \forall\ i,j \in J\ and\ i < j \tag{9}$$

$$r_{k} \geq (x_{i} + p_{i}) - Q\cdot\left(1 - y_{ik}^{B}\right), \quad for\ \forall i \in J;\ \forall k \in B \tag{10}$$

$$\sum_{n\in C} y_{kn}^{C} \leq 1, \quad for\ \forall k \in B \tag{11}$$

$$y_{kn}^{C} \geq R_{in}\cdot y_{ik}^{B}, \quad for\ \forall i \in J;\ \forall k \in B;\ \forall n \in C \tag{12}$$

$$r_{k} + \sum_{n\in C} h_{n}\cdot y_{kn}^{C} \leq r_{l} + Q\cdot\left(1 - \sum_{t\in T} z_{klt}^{T}\right), \quad for\ \forall k,l \in B;\ k \neq l \tag{13}$$

$$\sum_{t\in T} y_{kt}^{T} = 1, \quad for\ \forall k \in B \tag{14}$$

$$\sum_{k\in B} z_{kkt}^{T} \leq 1, \quad for\ \forall t \in T \tag{15}$$

$$\sum_{l\in B} z_{lkt}^{T} = y_{kt}^{T}, \quad for\ \forall k \in B;\ \forall t \in T \tag{16}$$

$$\sum_{\substack{l\in B \\ l\neq k}} z_{klt}^{T} \leq y_{kt}^{T}, \quad for\ \forall k \in B;\ \forall t \in T \tag{17}$$

$$r_{k} + \sum_{n\in C} h_{n}\cdot y_{kn}^{C} - d_{i} \leq \tau_{i} + Q\left(1 - y_{ik}^{B}\right), \quad for\ \forall i \in J;\ \forall k \in B \tag{18}$$

$$x_{i},\ \tau_{i} r_{k} \geq 0, \quad for\ \forall i \in J;\ \forall k \in B \tag{19}$$

$$y_{im}^{M},\ y_{ik}^{B},\ y_{kt}^{T},\ y_{kn}^{C} = 0\ or\ 1, \quad for\ \forall i \in J;\ \forall k \in B;\ \forall t \in T;\ \forall n \in C \tag{20}$$

$$z_{ijm}^{M},\ z_{klt}^{T} = 0\ or\ 1, \quad for\ \forall i,j \in J;\ \forall m \in M;\ \forall k,l \in B;\ \forall t \in T \tag{21}$$

Constraint (2) is to determine the precedence relation of producing jobs within the same machine at the manufacturing plant and calculate the starting time of processing the jobs. Constraint (3) restricts that each job must be assigned to one of the machines in the manufacturing plant. Constraints (4)–(6) have a relation that jobs assigned to the same machine must appear exactly once in their sequence. Constraint (4) ensures that the beginning of the production sequence in each machine can assign, at most, one job. So, one job must be assigned to the first position if the rest of jobs are succeeded by the job in each machine. Constraint (5) guarantees that one job will be immediately preceded by one job in a machine if it is assigned to the machine, and Constraint (6) also guarantees that one job will be immediately succeeded by at most one job, if the job is assigned to one of machines. Also, no succeeding job is allowed if the job exists at the last position of the sequence in machines. Constraint (7) ensures

that each job must be assigned to exactly one of the batches. Constraint (8) confirms that the total volume of jobs in batches must not be over the truck containing capacity.

Constraint (9) guarantees that all jobs in the same batch should belong and be shipped to the same customer. Constraint (10) restricts that the shipping starting time of each batch must be the longest completion time of manufacturing jobs in the batch. Constraints (11)–(12) force a relation between the jobs in the batch and the customer. In Constraint (13), the shipping time of each batch can be calculated by determining the precedence relation of the batches delivered by the same truck. Constraint (14) confirms that a truck must deliver only one batch to an associated customer at a time. Constraints (15)–(17) have a relation that batches assigned to the same truck must appear exactly once in their sequence. Constraint (15) ensures that the beginning of the delivery sequence in each truck can assign, at most, one batch. So, one batch must be assigned to the first position if the rest of the batches are succeeded by the batch in each truck. Constraint (5) guarantees that one batch will be immediately preceded by one batch in a truck if it is shipped to the truck, and Constraint (6) also guarantees that one batch will be immediately succeeded by at most one batch, if the batch is shipped to one of the trucks.

The above MILP formulation guarantees obtaining an optimal solution. However, the size of the formulation makes it hard to find an optimal solution within a limited time. This difficulty occurs from the number of integer variables and constraints. By the derived formulation, the numbers of integer variables and constraints depend on $\left( JM + BJ + TB + CB + J^2M + B^2T \right)$ and $\left\{ 4J + M + 4B + T + J^2 + B^2 + 3(JM + TB + BT) + JBC + BJ^2 + J^2M + B^2T \right\}$, respectively. Thus, CPLEX failed to obtain an optimal solution before running out of memory in large-sized problems. If the problem is reduced to consider only a machine scheduling problem, it is equivalent to a total tardiness parallel-machine scheduling problem. If the problem is reduced to considering only a batching problem, it is equivalent to a well-known bin-packing problem. If the problem is reduced to considering only a truck scheduling problem, it is equivalent to a total tardiness parallel-machine scheduling problem. Each of those problems are known to be –NP-hard [19]. Thus, it is necessary to propose an efficient heuristic to solve the problem within a short amount of time.

## 3. Variable Neighborhood Search (VNS) Algorithms

We develop VNS algorithms to solve the integrated scheduling problem efficiently. The VNS algorithm is designed to enrich the search space by restarting a local search heuristic with randomly generated neighborhood solutions from an incumbent solution by a pre-determined set of neighborhood structures. Systematic changes of the neighborhood solution within the local search is a key concept of VNS algorithms to improve a solution quality. Thus, the performance of VNS algorithm is mainly influenced by the local search heuristic and the neighborhood structure to meet problem characteristics [20].

The basic VNS algorithm starts with a randomly generated initial solution and repeats the shaking and moving steps until the termination condition (maximum neighborhood number) is met. In the shaking step, a neighbor of the incumbent solution is randomly generated, and the local search heuristic is performed. After the shaking step, the incumbent solution is compared with the local optimal solution and updates the incumbent solution when the local optimal solution is better. Algorithm 1 shows the procedure of the basic VNS scheme given as follows:

---

**Algorithm 1.** Basic VNS scheme

---

**Begin**
Find an initial solution set $S^*$.
Let iteration index $k \leftarrow 1$.
Define maximum neighborhood number $k_{max}$.
**While** $(k \leq k_{max})$
***Shaking***: find a random solution set $S \in N_k(S^*)$.
Perform a local search with $S$ to find a local optimum $S'$.
***Move or not***:
**If** $f(S') \leq f(S^*)$ **then**
$S^* \leftarrow S'$.
$k \leftarrow 1$.
**Else**
$k \leftarrow k + 1$.
**End If**
**End While**
**End**

---

*3.1. Neighboorhood Structure*

A solution of the integrated scheduling problem is represented by three sequences in this article: a job sequence for each machine, a job sequence for each batch, and a batch sequence for each truck. The example solution of the integrated schedule in Figure 1 is represented as shown in Figure 2.

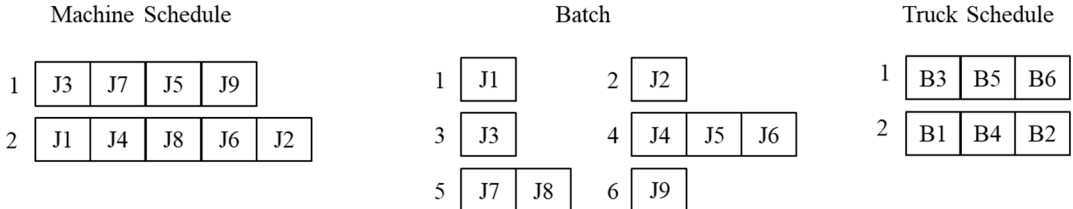

**Figure 2.** A solution representation for the integrated scheduling problem.

The neighborhood solutions are generated by operations describing how they change the incumbent solution. In this article, we use the following nine basic neighborhood operations, for machine schedule (M1, M2, M3, M4), batching (B1), and truck schedule (T1, T2, T3, T4):

3.1.1. Neighborhood Operations for Machine Schedule

- Within single machine: A machine and a job position $p$ in the job sequence of the machine are randomly selected. Another job position $q$ is randomly selected within the range [$max(0, p\text{-}gapJob)$, $min(p+gapJob$, current number of jobs assigned to the machine)]. The interval gap between selecting jobs is defined as $gapJob = n(J) \times preR$, where $preR$ ($0 \leq preR \leq 1$) is the predetermined relative interval ratio.

  M1.  InsertOnMachine (*gapJob*): The job at the position $p$ is removed and inserted to the position $q$ in the job sequence of the machine.
  M2.  SwapOnMachine (*gapJob*): The jobs at the positions $p$ and $q$ are interchanged in the job sequence of the machine.

- Across machines: A machine $m$ and a job position $p$ in the job sequence of the machine $m$ are randomly selected. Another machine $n$ and job position $q$ in the job sequence of the machine $n$ are randomly selected within the range [$max(0, p\text{-}gapJob)$, $min(p+gapJob$, current number of jobs assigned to the machine $n$)].

M3. *InsertAcrossMachine* (*gapJob*): The job at the position *p* in the job sequence of the machine *m* is removed and inserted to the position *q* in the job sequence of the machine *n*.

M4. SwapAcrossMachine (*gapJob*): The job at the positions *p* and *q* are interchanged in the associated job sequence of the machines *m* and *n*, respectively.

### 3.1.2. Neighborhood Operations for Batching

B. *SwapAcrossBatch*(): Two batches and job *i* and *j* in each batch are randomly selected with checking the feasibility of the truck capacity. The job *i* and *j* are interchanged.

### 3.1.3. Neighborhood Operations for Truck Schedule

- Within single truck: A truck and a batch position *r* in the batch sequence of the truck are randomly selected. Another batch position *s* is randomly selected within the range [*max*(0, *r-gapBatch*), min(*r+gapBatch*, current number of batches assigned to the truck)]. The interval gap between selecting batches is defined as *gapBatch* = *n(B)* × *preR*, where *preR* (0 ≤ *preR* ≤ 1) is the predetermined relative interval ratio.

  T1. InsertOnTruck (*gapBatch*): The batch in the position *r* is removed and inserted to the position *s* in the batch sequence of the truck.

  T2. SwapOnTruck (*gapBatch*): The batches in the position *r* and *s* are interchanged in the batch sequence of the truck.

- Across trucks: A truck *k* and a batch position *r* in the batch sequence of the truck *k* are randomly selected. Another truck *l* and batch position *s* in the batch sequence of the truck *l* are randomly selected within the range [*max*(0, *r-gapBatch*), *min*(*r+gapBatch*, current number of batches assigned to the truck *l*)].

  T3. InsertAcrossTruck (*gapBatch*): The batch at the position *r* in the batch sequence of the truck *k* is removed and inserted to the position *s* in the batch sequence of the truck *l*.

  T4. SwapAcrossTruck (*gapBatch*): The batch at the positions *r* and *s* are interchanged in the batch sequence of the associated trucks *k* and *l*, respectively.

In order to simultaneously determine machine schedule, batch assignment, and truck delivery schedule, a combination of the basic neighborhood operations for each decision would be examined in each iteration of the main VNS scheme. For the combination, the neighborhood operations for each decision are grouped, respectively, and an operation is randomly selected in each group when making a neighborhood. The grouped neighborhood structure is shown in Table 2.

**Table 2.** Grouped neighborhood structure.

| Group for Machine Schedule | Group for Batching | Group for Truck Schedule |
|:---:|:---:|:---:|
| {NONE, M1, M2, M3, M4} | {NONE, B} | {NONE, T1, T2, T3, T4} |

### *3.2. Local Search Scheme*

For the local search in VNS algorithms, we used a search method based on sequence arrays (SMSA). To apply SMSA to the integrated scheduling problem, three sequence arrays were used, which are machine scheduling, batching, and truck scheduling arrays. The sequence arrays, which are represented by a single dimensional string array with digits, and those require assignment rules to determine manufacturing sequence of machines, batching construction, and shipping sequence of trucks. The digits for machine, batching, and truck scheduling represent a job sequence to apply to the machine assignment rule, the batching rule, and the truck assignment rule, respectively [21]. The decoding processes of the sequence arrays are carried out by the three rules. Joo and Kim [16]

studied a similar problem (makespan problem) and compared the several assigning rules for their GA algorithm. The processing time and completion time-based assigning rules for machine and truck sequence arrays, and the minmax-based and rotation-based batching rules for batch sequence array were compared. They concluded that the completion time-based assigning rule for the machine and truck sequence arrays and the rotation-based batching rule for batch sequence array provided the best performance in terms of its effectiveness and efficiency for their algorithm. According to their result, we used three rules for decoding the sequence arrays to a compound solution for our local search scheme. The procedures of the three rules are as follows:

- **Machine assignment rule**: Calculate the completion times of each machine by temporarily assigning the current job to the end of the sequence in the corresponding machine. And then, find the machine with the shortest completion time is found and the job is permanently assigned to the machine and placed on the end-position of the manufacturing sequence of the machine.
- **Batching rule**: Find the first available batch, allowing the capacity of the batch to be greater than the volume of the job, as well as shipping towards the same customer and assigning the job to the batch**.** If no batch was satisfied by the conditions from the current available batches, create a new batch and assigned the job to the batch.
- **Truck assignment rule**: Calculate the completion times of each truck by temporarily assigning the current batch to the end of the sequence in the corresponding truck. And then, find the truck with the shortest completion time and the batch was permanently assigned to the truck and placed on the end-position of the shipping sequence of the truck.

For the local search with SMSA, we classified seven modification cases according to which sequence arrays were selected to apply the modification (see Table 3). Cases 1, 2, and 3 applied only one sequence array to the modification process and cases 4, 5, and 6 applied two of three sequence arrays to the modification process. Case 7 applies all three sequence arrays to the modification process. One of seven cases is randomly chosen according to the case selection probability $P_c$ in every iteration of the local search with SMSA.

**Table 3.** The modification cases.

| Case | Modify or Not | | | Case | Modify or Not | | |
|------|---------------------------|---------------------------|-------------------------|------|---------------------------|---------------------------|-------------------------|
|      | Machine Sequence Array | Batch Sequence Array | Truck Sequence Array |      | Machine Sequence Array | Batch Sequence Array | Truck Sequence Array |
| 1 | O | X | X | 4 | O | O | X |
| 2 | X | O | X | 5 | X | O | O |
| 3 | X | X | O | 6 | O | X | O |
|   |   |   |   | 7 | O | O | O |

In this article, we developed two kinds of procedures for the local search with SMSA. The difference between two procedures is to update whether the case selection probabilities are used or not when the modification of sequence array is processed. The first local search uses the static even case selection probability with the value 1/(the number of modification cases). The second local search uses the dynamic case selection probability, which is increased when the case is selected and decreases with a deterioration rate when the case is not selected. Three sequence operators for the modification of sequence array in SMSA were used. For the operators, two front and rear positions were randomly selected in the original sequence array.

- **Pull operator**: All digits between two positions (including the digits in the positions) are removed and placed to the end of the sequence array and the digits on the right side of the rear position are pulled to the position of the front point.

- **Insert operator**: The digit in the rear position is removed and simply inserted into the position in front of the digit in the front position.
- **Swap operator**: The two digits at the front position and the rear position are mutually interchanged.

The pseudo codes of two local search procedures with SMSA are given in Algorithm 2 and 3 as follows:

---

**Algorithm 2.** Local search with *static case selection probability*

---

**Begin**
Define termination count $t_{max}$.
Define number of modification cases $C_{max}$
Let the initial case selection probability $P_c \leftarrow 1/C_{max}$.
Find an initial sequence arrays $C^*$ encoded from the solution set $S$.
Let current local optimum $S' \leftarrow S$.
Let iteration index $t \leftarrow 1$.
**While** $(t \leq t_{max})$
*Modification of Sequence Arrays*:
Find a random number $r \leftarrow uniform(0,1)$.
Let the case index $c \leftarrow 1$.
**While** $(c \leq C_{max})$
**If** $r \leq \sum_{j=1}^{c} P_j$ **then**
**Selecting case** $cs \leftarrow c$.
**Break**
**End If**
**End While**
Modify the sequence arrays $C^*$ to a new sequence arrays $C$ according to case $cs$.
Generate a corresponding solution set $S''$ decoded from the sequence arrays $C$.
*Move or not*:
**If** $f(S'') \leq f(S')$ **then**
$S' \leftarrow S''$.
$C^* \leftarrow C$.
$t \leftarrow 1$.
**Else**
$t \leftarrow t + 1$.
**End If**
**End While**
**Return** the local optimum $S'$.
**End**

---

---

**Algorithm 3.** Local search with *dynamic case selection probability*

---

**Begin**
Define termination count $t_{max}$.
Define number of modification cases $C_{max}$
Let the initial case selection probability $P_c \leftarrow 1/C_{max}$.
Let the deterioration rate $\alpha, \ 0 < \alpha < 1$.
Find an initial sequence arrays $C^*$ encoded from the solution set $S$.
Let current local optimum $S' \leftarrow S$.
Let iteration index $t \leftarrow 1$.
**While** $(t \leq t_{max})$
*Modification of Sequence Arrays*:
Find a random number $r \leftarrow uniform(0,1)$.
Let the case index $c \leftarrow 1$.
**While** $(c \leq C_{max})$
**If** $r \leq \sum_{j=1}^{c} P_j$ **then**
**Selecting case** $cs \leftarrow c$.
**Break**
**End If**
**End While**
Modify the sequence arrays $C^*$ to a new sequence arrays $C$ according to case $cs$.
Generate a corresponding solution set $S''$ decoded from the sequence arrays $C$.
*Move or not*:
**If** $f(S'') \leq f(S')$ **then**
$P_{cs} \leftarrow P_{cs} + \{f(S') - f(S'')\}/f(S')$.
$S' \leftarrow S''$.
$C^* \leftarrow C$.
$t \leftarrow 1$.
**Else**
$P_{cs} \leftarrow P_{cs} \times \alpha$.
$t \leftarrow t + 1$.
**End If**
**End While**
**Return** the local optimum $S'$.
**End**

---

### 3.3. Encoding and Decoding of a Solution

SMSA used for local search in VNS algorithms is operated with three sequence arrays that have single dimensional string arrays. So, the encoding and decoding procedures between the compound (integrated) solution and sequence arrays for SMSA are required. The procedures used are the three assigning rules applied to the local search in Section 3.2. Figure 3 describes encoding and decoding procedures with the example integrated schedule presented in Figure 2 and Table 1.

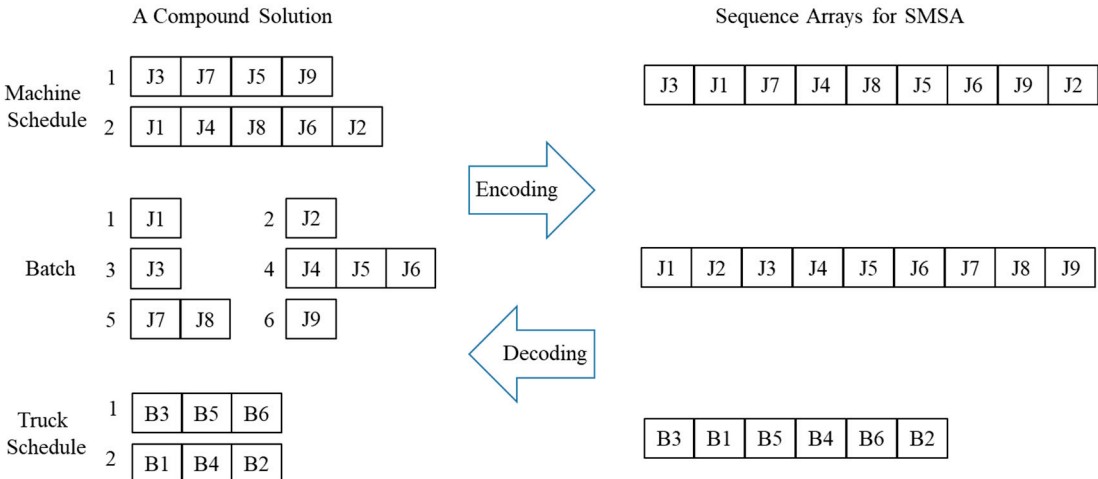

**Figure 3.** Encoding and decoding procedures.

## 4. Computational Testing Experiments

In this section, we conduct extensive computational testing experiments to access the performance of two VNS algorithms using the local search with static case selection probability (VNS_S) and VNS algorithm using the local search with dynamic case selection probability (VNS_D).

Since the problem complexity increases as the number of jobs (*J*) increases, we divided the test problems into two groups using *J*. The first group is to compare the performances of VNS algorithms with the optimal solution, and the test problems in the group are randomly generated with *J* selecting between 5 and 10. For the optimal solution, ILOG CPLEX 12.7 was adopted to solve the MILP formulation in Section 2. We terminated a particular run if an optimal solution was not found in an imposed 7200 (s) time limit. The test problems in the second group are randomly generated with *J* selecting more than 10. The group is to compare the relative performance of the solutions obtained by VNS algorithms. VNS algorithms were coded with the language C#, and all experiments were tested on a PC with 1.86 GHz Intel Core 2 CPU processor and 2 GB RAM.

The problem complexity is affected by five problem parameters which are the number of jobs (*J*), the number of machines (*M*), the number of trucks (*T*), the number of customers (*C*), and tardiness factor ($\delta$)). Test problems are randomly made according to the five parameters. The tardiness factor $\delta$ ($0 \leq \delta \leq 1$) is to generate the due times for each job. The due times for each job generated are more scattered as the value of $\delta$ is increased. Three values, 0.1, 0.3, and 0.5, are considered for the tardiness factor.

In the small-sized group, eight test problems were randomly generated for each tardiness factor. Under the predetermined one of tardiness factor values, the size of four problem parameters, which are the numbers of jobs, machines, trucks, and customers, were randomly selected by U [5,10], U [2,6], U [2,4], and U [3,4], respectively. In the large-sized group, a total of 24 test problems with 20, 40, and 60 jobs were randomly generated for each tardiness factor. Under the predetermined one of job sizes and one of tardiness factor values, the size of three problem parameters, which are the numbers of machines, trucks, and customers, were randomly selected by U [3,6], U [2,4], and U [3,6], respectively. The processing time and the transportation time were randomly selected by U [60,120] and U [60,240]. The volume of jobs was randomly selected by U [5,10] with a fixed value of the batch capacity as 20. The relative interval ratio selected by truck schedule were predetermined as {0.05, 0.1, 0.2, 0.4, 0.7, 1.0}.

The performance of VNS algorithms were relatively compared, and two performance measures, called relative deviation index (RDI) and mean absolute deviation (MAD), were defined. The measures are expressed by Equations (22) and (23), respectively.

$$\mathrm{RDI} = \frac{OBJ_{\mathrm{sol}} - OBJ_{\mathrm{best}}}{OBJ_{\mathrm{worst}} - OBJ_{\mathrm{best}}}, \tag{22}$$

where $OBJ_{\mathrm{best}}$ and $OBJ_{\mathrm{worst}}$ are the objective function values of the best feasible solution obtained by one of the algorithms (or the optimal solution by CPLEX) and the worst feasible solution obtained by one of the algorithms, respectively. $OBJ_{\mathrm{sol}}$ is an objective function value obtained by any VNS algorithm.

$$\mathrm{MAD}(\%) = \frac{|OBJ_{sol} - OBJ_{\mathrm{mean}}|}{OBJ_{\mathrm{mean}}} \times 100, \tag{23}$$

where $OBJ_{\mathrm{mean}}$ is a mean value of the replicated objective function values by any VNS algorithm.

All problems were tested by 30 replications. The performance results of the small-sized group are presented in Table 4. The objective function values of the optimal solution by CPLEX are represented, and the average RDI and MAD by VNS algorithms are compared. For the small-sized group, low values of RDI and MAD are indications that all VNS algorithms give good performances. In Table 4, computational times (CPU times) of instances are also calculated. We can find the CPU time of CPLEX exponentially increases as the number of jobs increases. Meanwhile, CPLEX could not search an optimal solution for problems over 7–8 jobs in the given time limit.

**Table 4.** Test results of small-sized group.

| Test Problems | | | | | CPLEX | | VNS_S | | | VNS_D | | |
|---|---|---|---|---|---|---|---|---|---|---|---|---|
| $\delta$ | $J$ | $M$ | $T$ | $C$ | Obj. | CPU | RDI | MAD | CPU | RDI | MAD | CPU |
| 0.1 | 5 | 3 | 2 | 4 | 918 | 2.52 | 0.00 | 0.00 | 0.02 | 0.00 | 0.00 | 0.05 |
| | 5 | 4 | 3 | 3 | 1068 | 1.55 | 1.00 | 0.00 | 0.02 | 1.00 | 0.00 | 0.05 |
| | 6 | 5 | 3 | 2 | 896 | 118.18 | 1.00 | 0.00 | 0.03 | 1.00 | 0.00 | 0.06 |
| | 6 | 3 | 2 | 5 | 1306 | 217.10 | 0.00 | 0.00 | 0.03 | 0.00 | 0.00 | 0.11 |
| | 7 | 3 | 4 | 3 | 1994 | 7200+ | 0.00 | 0.00 | 0.05 | 0.00 | 0.00 | 0.16 |
| | 8 | 5 | 2 | 5 | 2007 | 7200+ | 0.00 | 0.00 | 0.06 | 0.00 | 0.00 | 0.16 |
| | 9 | 4 | 4 | 6 | 2091 | 7200+ | 0.86 | 0.10 | 0.08 | 0.89 | 0.23 | 0.13 |
| | 10 | 6 | 3 | 4 | 2717 | 7200+ | 0.00 | 0.00 | 0.08 | 0.00 | 0.00 | 0.21 |
| 0.3 | 5 | 3 | 2 | 4 | 371 | 2.36 | 0.25 | 0.00 | 0.02 | 0.25 | 0.00 | 0.07 |
| | 5 | 4 | 2 | 3 | 720 | 2.08 | 0.00 | 0.00 | 0.01 | 0.00 | 0.00 | 0.04 |
| | 6 | 5 | 4 | 2 | 757 | 1.78 | 0.00 | 0.00 | 0.03 | 0.00 | 0.00 | 0.08 |
| | 6 | 3 | 2 | 5 | 674 | 25.83 | 0.00 | 0.00 | 0.02 | 0.00 | 0.00 | 0.10 |
| | 7 | 3 | 3 | 3 | 308 | 7200+ | 0.25 | 0.00 | 0.04 | 0.25 | 0.00 | 0.14 |
| | 8 | 5 | 3 | 5 | 787 | 7200+ | 0.00 | 0.00 | 0.06 | 0.00 | 0.00 | 0.16 |
| | 9 | 4 | 4 | 6 | 439 | 7200+ | 0.33 | 0.00 | 0.08 | 0.33 | 0.00 | 0.17 |
| | 10 | 6 | 3 | 4 | 1489 | 7200+ | 0.00 | 0.00 | 0.08 | 0.00 | 0.00 | 0.21 |
| 0.5 | 5 | 3 | 3 | 3 | 42 | 1.11 | 1.00 | 0.00 | 0.01 | 1.00 | 0.00 | 0.04 |
| | 5 | 4 | 3 | 4 | 298 | 1.69 | 1.00 | 0.00 | 0.02 | 1.00 | 0.00 | 0.03 |
| | 6 | 5 | 2 | 2 | 0 | 0.44 | 0.00 | 0.00 | 0.02 | 0.00 | 0.00 | 0.05 |
| | 6 | 3 | 4 | 5 | 442 | 133.63 | 0.78 | 0.00 | 0.03 | 0.78 | 0.00 | 0.07 |
| | 7 | 5 | 3 | 5 | 369 | 102.64 | 0.00 | 0.00 | 0.04 | 0.00 | 0.00 | 0.09 |
| | 8 | 6 | 2 | 6 | 64 | 7200+ | 0.00 | 0.00 | 0.05 | 0.00 | 0.00 | 0.14 |
| | 9 | 4 | 4 | 3 | 138 | 7200+ | 0.21 | 0.13 | 0.08 | 0.20 | 0.00 | 0.24 |
| | 10 | 6 | 2 | 6 | 143 | 7200+ | 0.00 | 0.00 | 0.08 | 0.00 | 0.00 | 0.18 |
| | Average | | | | | | 0.28 | 0.01 | 0.04 | 0.28 | 0.01 | 0.11 |

In Table 5, the performance results of the large-sized group are summarized. The average RDI and MAD of VNS_D is lower than those of VNS_S. The results indicate that VNS algorithms with a local search with dynamic case selection probability significantly improve the performance of the algorithms compared to VNS algorithms with a local search with static case selection probability. The CPU times of each VNS algorithm are short enough to obtain the best solution. The observed differences between two local search schemes are more statistically significant as tardiness factors decrease. This result

indicates that VNS_D gives a better performance for the proposed scheduling problem as the due date becomes more tightly controlled.

**Table 5.** Test results of large-sized group.

| Test Problems | | VNS_S | | | VNS_D | | | GA | | |
|---|---|---|---|---|---|---|---|---|---|---|
| δ | J | RDI | MAD | CPU | RDI | MAD | CPU | RDI | MAD | CPU |
| 0.1 | 20 | 0.14 | 0.76 | 1.09 | 0.08 | 0.64 | 2.50 | 0.60 | 2.25 | 1.51 |
| | 40 | 0.18 | 1.18 | 9.51 | 0.09 | 0.91 | 15.07 | 0.70 | 2.68 | 8.63 |
| | 60 | 0.25 | 1.88 | 37.45 | 0.12 | 1.42 | 55.08 | 0.79 | 2.48 | 24.16 |
| | 80 | 0.15 | 11.44 | 78.75 | 0.05 | 8.79 | 102.96 | 0.85 | 5.94 | 37.43 |
| | 100 | 0.08 | 19.77 | 200.82 | 0.05 | 27.14 | 195.62 | 0.86 | 7.27 | 67.41 |
| 0.3 | 20 | 0.14 | 2.51 | 6.92 | 0.08 | 1.84 | 11.34 | 0.59 | 6.18 | 1.43 |
| | 40 | 0.24 | 5.14 | 6.67 | 0.14 | 3.76 | 9.90 | 0.77 | 5.64 | 7.20 |
| | 60 | 0.22 | 8.25 | 22.32 | 0.12 | 9.92 | 27.40 | 0.73 | 9.93 | 20.31 |
| | 80 | 0.07 | 41.01 | 74.15 | 0.02 | 57.11 | 83.61 | 0.84 | 9.51 | 36.95 |
| | 100 | 0.05 | 38.68 | 166.17 | 0.03 | 32.83 | 101.86 | 0.83 | 10.20 | 67.53 |
| 0.5 | 20 | 0.04 | 38.78 | 3.06 | 0.05 | 23.04 | 3.18 | 0.31 | 38.74 | 1.36 |
| | 40 | 0.01 | 4.76 | 1.17 | 0.01 | 8.78 | 1.37 | 0.00 | 0.00 | 7.21 |
| | 60 | 0.03 | 1.88 | 4.00 | 0.02 | 2.29 | 4.08 | 0.12 | 10.75 | 18.70 |
| | 80 | 0.04 | 27.80 | 50.01 | 0.03 | 33.98 | 40.95 | 0.76 | 16.56 | 36.80 |
| | 100 | 0.03 | 55.51 | 111.13 | 0.02 | 21.86 | 69.68 | 0.72 | 18.65 | 66.95 |
| Average | | 0.11 | 17.29 | 51.55 | 0.06 | 15.62 | 48.31 | 0.64 | 9.83 | 26.92 |

To compare the performances of the VNS algorithms with the other meta-heuristics, we tested GA-based algorithms with the same problem sets. The performance results are also presented in Table 5. The tested GA is a single-stage algorithm with independent dispatching rules. It uses a chromosome representing two string arrays for machine and truck scheduling and one string array with job batching. In this article, VNS algorithms proposed give better performance than GA in any job size and any tardiness factors. The results indicate that VNS algorithms improve the performance by broadly exploring the solution space compared with GA.

## 5. Conclusions

This article considered an integrated problem of one batching and two scheduling decisions between a manufacturing plant and multi-delivery sites. Many jobs ordered by multiple customers are firstly manufactured by one of machines in the plant. In this problem, two scheduling (machine and delivery truck scheduling) problems and one batching problem must be simultaneously determined to minimize the total tardiness. To find the optimal solution, a MILP model was developed. We mainly found an optimal solution using CPLEX for small-sized groups, but it was inefficient and impractical to find the optimal solution for the problems of large-sized groups. Thus, two VNS algorithms, which were applied with different local search schemes, were applied to improve the performance of the algorithm. We conclude that the VNS algorithm with dynamic case selection probability finds better solutions in reasonable CPU times, compared with the VNS algorithm with static case selection probability and the GA based on the test results.

**Author Contributions:** Conceptualization, B.S.K.; methodology, C.M.J. and B.S.K.; software, C.M.J. and B.S.K.; validation, C.M.J.; formal analysis, C.M.J. and B.S.K.; investigation, B.S.K.; data curation, B.S.K.; writing—original draft preparation, C.M.J. and B.S.K.; writing—review and editing, C.M.J. and B.S.K.; visualization, C.M.J.; funding acquisition, B.S.K.

**Funding:** This work was supported by the Incheon National University Research Grant in 2017.

**Conflicts of Interest:** The authors declare no conflict of interest.

## Appendix A

For the mathematical formulation, the parameters and decision variables were defined.

### *Parameters*

| | |
|---|---|
| $J$ : | job-set |
| M: | machine-set |
| $B$ : | batch-set |
| $T$ : | truck-set |
| $C$ : | customer-set |
| $p_j$ : | processing time without deterioration of job $j \in J$ |
| $d_j$ : | due time of job $j \in J$ |
| $h_n$ : | transportation time, including return time for delivery of customer $n \in C$ |
| $R_{jn}$ : | 1, if job $j \in J$ is required by customer $n \in C$; 0 otherwise |
| $v_j$ : | volume of job $j \in J$ |
| $V$ : | truck containing capacity |
| Q: | A large value |

### *Continuous variables*

| | |
|---|---|
| $x_j$ : | production starting time of job $j$ |
| $r_k$ : | shipping starting time of batch $k$ |
| $\tau_j$ : | tardiness of job $j$ |

### *Binary variables*

| | |
|---|---|
| $y_{im}^M$ : | 1 if machine $m$ assigns job $i$ at the manufacturing plant; 0 otherwise |
| $y_{ik}^B$ : | 1 if batch $k$ assigns job $i$; 0 otherwise |
| $y_{kt}^T$ : | 1 if truck $t$ assigns batch $k$; 0 otherwise |
| $y_{km}^C$ : | 1 if customer $m$ assigns batch $k$; 0 otherwise |
| $z_{ijm}^M$ : | 1 if job $i$ immediately precedes job $j$ at machine $m$ at the manufacturing plant; 0 otherwise |
| $z_{klt}^T$ : | 1 if batch $k$ immediately precedes batch $l$ in truck $t$; 0 otherwise |

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
