# Peer review of "Variable Neighborhood Search Algorithms for an Integrated Manufacturing and Batch Delivery Scheduling Minimizing Total Tardiness"

_applsci, doi:10.3390/app9214702_

Round 1
Reviewer 1 Report
The technical writing of this paper is good. The authors gave a couple of simple examples to illustrate their algorithm properly. However, I do have a major concern of the scalability of the proposed algorithms stated in my first comment. Other comments are minor for paper improvement.
The methods proposed in this paper are Integer Linear Programming (ILP) based ones, and the solver CPLEX has been used to give the optimized solution. It is well known that ILP has the scalability issue that solving large scale problem may take very long time. Then, it might not be proper for this industry problems, as the solution needs to be made in short time. The maximum sized group used in the evaluation is only 60, which may not match the industrial needs. I strongly suggest the authors use the real data from the industry to evaluate their methods. The conclusion needs to involved into the abstract. It says the proposed methods has been compared, but what is the result and conclusion. What is 3PL? It needs a short explain here. For the processing time a job Pj, how can you get the accurate estimated value? The author mentioned "Sometimes, no jobs can be assigned to some batches in the set B". Why? For the predefined neighborhoods, it only considers single job/batch swap/insert senarios. How about multiple job/batch cases?Author Response
Please see the attachment

Reviewer 2 Report
The authors proposed a VNS algorithm to solve the integrated manufacturing and delivery problem with parallel manufacturing. The algorithm is novel with noticeably accelerated computational time compared to CPLEX and improved stability compared to GA. There are a few minor comments for the paper:
What is the time complexity of the algorithm regarding size of J, M, B, T and C?
Can the algorithm be extended to parallel when the scale of problem become huge? What is the size of problems in practical applications?
For Table 4, since reference value from CPLEX is available, it would be better to calculate relative error to CPLEX besides RDI and MAD.
Reviewer 3 Report
The paper focuses on supply chain management (SCM), and discusses an integrated problem of one batch and two schedules (machine and truck) decision making between a manufacturing plant and multi-delivery sites. This goal of this paper tried to present a solution to integrate parallel manufacturing products, grouping to the same place, and deliver to multiples corresponding customers within a limited capacity such as the total tardiness of jobs is minimized. The authors proposed two effective and efficient variable neighborhood search (VNS) algorithms for minimizing total tardiness of our integrated scheduling framework.
Reviewer 4 Report
The paper is general well written and the approach is correct.
The validation of the approach through simulations based on randomly generated data is in favour of the proposal.
Minor corrections are suggested:
(1) do not use acronyms in the abstract; modify the second part of the 5th sentence of the abstract (after and there is no predicate); "randomly generated problems" is not in favor of highlighting the result -- please reformulate (eg. how many tests)
(2) subsection 3.1.2: there is and B1 as item but no B2 -- please remove B1
(3) Section 4, "the test problems in the group are randomly generated within 10 jobs." - within? please reformulate/clarify
Round 2
Reviewer 1 Report
I am satisfied with the response and revision from the authors, and very appreciate their efforts of addressing my comments.Author Response
Please see the attachment. It include a replying comment for the addtional minor review of the reviewer 3.
